# Osteopathic practice in the United Kingdom: A retrospective analysis of practice data

**Austin Plunkett** [1]*, **Carol Fawkes**[1,2], **Dawn Carnes**[1,2]

**1** Institute of Population Health Sciences, Barts and the London School of Medicine and Dentistry, London, United Kingdom, **2** National Council for Osteopathic Research. Membership is described at https://ncor.org.uk

☉ These authors contributed equally to this work.
\* a.plunkett@qmul.ac.uk

## Abstract

### Background

This study describes osteopathic practise activity, scope of practice and the osteopathic patient profile in order to understand the role osteopathy plays within the United Kingdom's (UK) health system a decade after our previous survey.

### Method

We used a retrospective questionnaire survey design to ask about osteopathic practice and audit patient case notes. All UK registered osteopaths were invited to participate in the survey. The survey was conducted using a web-based system. Each participating osteopath was asked about themselves, their practice and asked to randomly select and extract data from up to 8 random new patient health records during 2018. All patient related data were anonymised.

### Results

The survey response rate was 500 osteopaths (9.4% of the profession) who provided information about 395 patients and 2,215 consultations. Most osteopaths were self-employed (81.1%; 344/424 responses) working alone either exclusively or often (63.9%; 237/371) and were able to offer 48.6% of patients an appointment within 3 days (184/379). Patient ages ranged from 1 month to 96 years (mean 44.7 years, Std Dev. 21.5), of these 58.4% (227/389) were female. Infants <1 years old represented 4.8% (18/379) of patients. The majority of patients presented with musculoskeletal complaints (81.0%; 306/378). Persistent complaints (present for more than 12 weeks before appointment) were the most common (67.9%; 256/377) and 41.7% (156/374) of patients had co-existing medical conditions. The most common treatment approaches used at the first appointment were soft-tissue techniques (73.9%; 292/395), articulatory techniques (69.4%; 274/395) and high velocity low amplitude thrust (34.4%; 136/395). The mean number of treatments per patient was 7 (mode 4).

**Data Availability Statement:** The dataset underlying all results is available in a permanent public registry at DOI:10.17605/OSF.IO/A25XW. Identifying or personal aspects of the data are not shared due to participant confidentiality, and

remain the property of the United Kingdom's National Council for Osteopathic Research, an independent charity. Requests for data by researchers who meet the criteria for access to confidential data may be made by contacting NCOR via www.ncor.org.uk.

**Funding:** The study was funded by the UK's National Council for Osteopathic Research (NCOR), registered UK charity number 1157217. AP and CF were employed by NCOR and DC was Director of NCOR. Website for NCOR: https://ncor.org.uk.

**Competing interests:** AP (corresponding author) is a Director at the Institute of Osteopathy, the members' organisation for the UK's osteopathic profession. We confirm that this does not alter our adherence to all PLOS ONE policies on sharing data and materials, and does not influence our sharing of data and/or materials.

## Conclusion

Osteopaths predominantly provide care of musculoskeletal conditions, typically in private practice. To better understand the role of osteopathy in UK health service delivery, the profession needs to do more research with patients in order to understand their needs and their expected outcomes of care, and for this to inform osteopathic practice and education.

## Introduction

Osteopathy has formed part of the provision of regulated musculoskeletal services in the United Kingdom (UK) for almost three decades and features as part of both national and international clinical guidelines [1–3]. Although osteopathy is perhaps most recognised for its use of spinal manipulation, osteopathic practise encompasses a range of manual therapy techniques appropriate to individual patients, and includes also education and advice [4].

In 2009 the UK's National Council for Osteopathic Research (NCOR) conducted a survey of osteopaths to describe the full extent of their practise and the patient population consulting for osteopathic care [4] The survey was based on a standardised data collection (SDC) tool developed by practising osteopaths [5].

The previous 2009 UK study indicated that the majority of patients sought osteopathic care for low back pain (36%) and neck and shoulder pain (21.8%). The majority were female (56%) and patients ranged between the ages of 0–93 years [4]. This information was useful to understand the actual role and the potential role osteopathy could play in UK health care provision. Since 2009, we anticipated that the role of osteopaths in the UK would have changed in line with infrastructure changes within the provision of national healthcare. In 2012 the UK Government changed the way in which healthcare provision is commissioned [6] with the introduction of a new low back pain pathway and the creation of the First Contact Practitioner role, both of which provided opportunities for osteopaths to work as part of multidisciplinary teams. Additionally, in 2017 osteopaths became recognised as part of the Allied Health Professions (AHP) community which has enhanced potential roles for osteopaths within the UK National Health Service (NHS) [7].

Information from the 2009 SDC study has been used extensively to describe the full extent of osteopathic practise in the UK, Subsequently, similar work has been undertaken in Belgium, Spain, Switzerland and Italy to describe osteopathic care in those countries [8–11]. This paper updates the 2009 UK study and contributes to the emerging literature describing current osteopathic practises across Europe.

### Aim

The aim of this study is to provide the osteopathic community, patients, the public, other health care professionals and policy-makers with a descriptive profile of osteopathic practice, the osteopathic patient population and the care they receive from osteopaths. This study will help to formulate teaching goals, plan ongoing continuing professional development activities, identify national research priorities, provide data for stakeholder negotiation and policy decision-making and ultimately to optimise patient care.

## Method

### Design

We used a retrospective questionnaire survey design to: i) ask about osteopaths and their osteopathic practice and ii) audit patient case notes. The survey was a practice review, a type of service evaluation using the principles of audit [12]. The retrospective design meant that we were evaluating actual recorded data, therefore some data may be missing in patient records. This type of design can help to understand actual practice as reflected by the record keeping of the osteopaths.

### Population and setting

All osteopaths in the UK are required to be registered with the General Osteopathic Council (GOsC), a statutory body set up for patient protection. Osteopaths must undergo training at a recognised osteopathic education institution or be trained to an equivalent standard elsewhere so that the practicing osteopath is capable and able to comply with the Osteopathic Practice Standards [13]. All registered osteopaths were invited to participate in the study. There were 5,341 registered osteopaths in 2019 (www.osteopathy.org.uk). Only information about patients and osteopaths in the private health sector setting was sought for this study. A very small proportion of osteopaths work in the National Health Service (NHS), data about and from the NHS was not collected as part of this study.

### The questionnaire survey

Survey questions were selected from the SDC survey which had been used successfully in 2009 [4] and from a survey commissioned by the GOsC [14]. Four additional questions were added regarding osteopaths' engagement with sources of evidence. Face validity of these questions was confirmed with stakeholders and by piloting with a small group of osteopaths. The full questionnaire consisted of three parts, described in S1 File. Part A contained information about the survey and asked for consent to participate. Part B asked about the osteopath respondent. Part C asked about osteopathic patients and practice, based on data collected through review of patient records from the year 2018.

Study data were collected and managed using the Research Electronic Data Capture (REDCap) software, a secure, web-based application designed to support data capture for research studies [15].

### Recruitment

All GOsC osteopaths registered as practicing were invited to participate in this online-survey. They were invited by email by the GOsC at the request of the National Council for Osteopathic Research (NCOR), the funding body and research team, as well as by emails directly from NCOR. Recruitment commenced in mid-July 2019 and the survey closed at the end of January 2020. During this period the survey was promoted on social media (Facebook and Twitter) and in the magazines of the osteopathic professions' regulator and membership organisation (*The Osteopath* and *Osteopathy Today* respectively) to promote the survey and encourage participation. Osteopaths were informed that they could use their participation in the survey as part of their continuing professional development (CPD).

The REDCap survey platform allocated a random ID to participants so they could return to the survey and continue later. Those who wished to participate in the survey were asked to provide their consent online before commencing the survey. Once section A was completed, the system automatically registered them as participants and opened section B of the survey.

We also asked osteopaths who did not want to participate their reasons for non-participation where relevant. Reminders were sent by email after one month and two months.

## Health record sample selection

We required the osteopaths to give us data about new patients throughout the year in 2018. Selecting patients from both new and returning encounters may lead to an over-representation of those consulting more frequently (i.e. those seeking care more often are more likely to be selected), therefore we decided to profile only new patients.

To select records, we provided each osteopaths with a random date from 2018, generated by a third-party provider of true random data [16]. Participants were instructed to find the first new patient on or after the provided random date.

## Anonymity

**Osteopath anonymity.** All participants were provided with a unique identifier for use when returning to the survey. The survey database was only consulted where an osteopath forgot or lost their study identifier number. In order to ensure that no unique combination of data could be used to identify any individual osteopath, personal data was collected in ranges. For example, age-ranges were collected rather than ages, and data regarding years in practice was collected in 2-year ranges.

Section B of the questionnaire was not linked in any way to section C, thereby reassuring participants that their responses regarding patient care and management could not be used to identify them.

**Patient anonymity.** The separation of part B from part C contributed to ensuring patient anonymity. Directly identifiable patient data was not collected in order to ensure patient anonymity. Osteopaths were asked not to include records where a patient's health might be an identifying factor, e.g. very rare disease. All data was combined and analysed, no individual information is presented in isolation as a case.

## Pilot testing

To assure external validity we asked osteopaths, stakeholders and researchers (10 people in total) to comment on and test the questionnaire's face and content validity. For internal validity, we pre-tested the software for reliability of health record selection, data entry and data extraction.

## Sampling and sample size

In the previous survey of osteopaths during 2009, a 9.4% response rate was achieved: 342 osteopaths participated contributing data about 1,630 patients. For 2019, a representative sample of osteopaths was estimated at 359 from 5,341 registered osteopaths using a confidence interval of 95% with a 5% margin of error). Using a 10% response rate a minimum of 3,590 osteopaths needed to be contacted. However, for the sake of inclusiveness all registered osteopaths were invited to take part as we were asking osteopaths to review fewer patient records than the last data collection exercise (up to 8, whereas in the previous survey we asked for 10).

## Statistical analysis

Descriptive statistics were used to describe both the osteopaths, osteopathic patients and osteopathic practice. Statistical analysis was conducted using the reporting functionality built into REDCap where possible. Where this was not possible OpenOffice's *LibreOffice Calc* and the

Python programming language were used. Continuous variables are presented where feasible as means with standard deviations. Categorical data are presented as frequencies with percentages. Percentages were rounded to two decimal places.

Both fully-completed and partially-completed patient records were included for analysis. Consequently, patient-related statistics have variable total responses.

Data describing the demographic characteristics of the UK's register of osteopaths was obtained from the General Osteopathic Council (GOsC) to determine representativeness.

## Managing missing data

For data extraction from the health records, respondents were given the opportunity to answer 'don't know/can't tell from records'. For other questions, osteopaths were permitted to leave an entry blank and provide a text for explanation. Partial data occurred when a participant stopped answering the survey questions before completion.

## Ethics and governance

The study protocol was reviewed and written approval was provided by the Queen Mary, University of London Ethics of Research Committee Panel D, reference "QMERC2019/23", on 23rd May 2019.

## Informed consent from participating osteopaths

All participating osteopaths were asked to read the information about the study and provide their consent in the first section of the survey prior to engaging in the study.

## Data security and protection

Data protection was guaranteed at the level of data handling and data hosting via the firewalled university servers, and was encrypted in transit over the Internet. Data was entirely anonymous and IP addresses were not made available. The full dataset was only made accessible to the study staff and the staff responsible for the survey software.

All was handled in accordance General Data Protection Regulation laws and guidance set therein, anonymised and used in accordance with the guidance set out in Health and Social Care Act 2012 on Good Clinical Practice in research.

# Results

## Survey participation data

During 2019, all 5,341 osteopaths registered with the GOsC were invited to participate in the survey. 500 osteopaths provided data for analysis, representing 9.4% of registered osteopaths. They contributed information about 395 patients and 2,215 consultations.

The most frequent age-range for respondents was 51–55 years old (22.9%; 95/415). The median age fell within the age-group 46–50 years. Females represented 59.1% (n = 254) and 98.6% (423/429) gained their qualification to practice in the UK. The median 'years in practice' fell in the range 19–20 years in practice.

## Practice data

The number of patients seen during the week (Monday to Friday) varied from 2 to 105 with modes of 20 and 30. Nearly two-thirds (63.3%) of osteopaths did not see patients at weekends. The number of *new* patients seen throughout the week varied from 0 to 80 with a mean of 7.

Just under half of patient appointments (48.6%; 184/379) were available within 3 days, with only 6.9% (26/379) of appointment waiting times being longer than a week. The most commonly-experienced waiting time was 2–3 days (33.0%; 125/379).

Most patients paid for their appointments themselves (88.4%; 327/370).

The majority (81.1%) of osteopaths (344/424) were self-employed. There were 12.7% associate osteopaths who did not have a contract of employment (54/424) and 4.3% with a contract of employment (18/424).

Most osteopaths (63.9%; 237/371) worked alone often or exclusively.

## Patient characteristics

**Age and gender of patients.** More females than males sought osteopathic care 58.4% vs 41.6% (227 vs 167 records).

The age profile of patients showed that 53.8% of patients were between 30 and 60 years old. Nearly 10% were under 10 years old and of these 4.8% were under 1 years old. Fig 1 shows the age profile of patients.

**Previous experience of osteopathy.** Over half of patients had not seen an osteopath before (57.7%; 226/392). Of those who *had* seen an osteopath before, just over half (51.4%; 75/146) had seen a different osteopath previously.

**Presenting complaint.** The patient's main presenting complaints were musculoskeletal pain or dysfunction (81%; 306/378). Full data are presented in Table 1.

**Co-existing conditions.** 41.7% (156/374) of patients had current co-existing conditions diagnosed by a medical practitioner. The most common co-existing conditions were: hypertension ($n = 41$); arthritis ($n = 31$); anxiety($n = 22$); asthma ($n = 19$); migraine ($n = 16$); diabetes ($n = 14$); irritable bowel syndrome ($n = 13$).

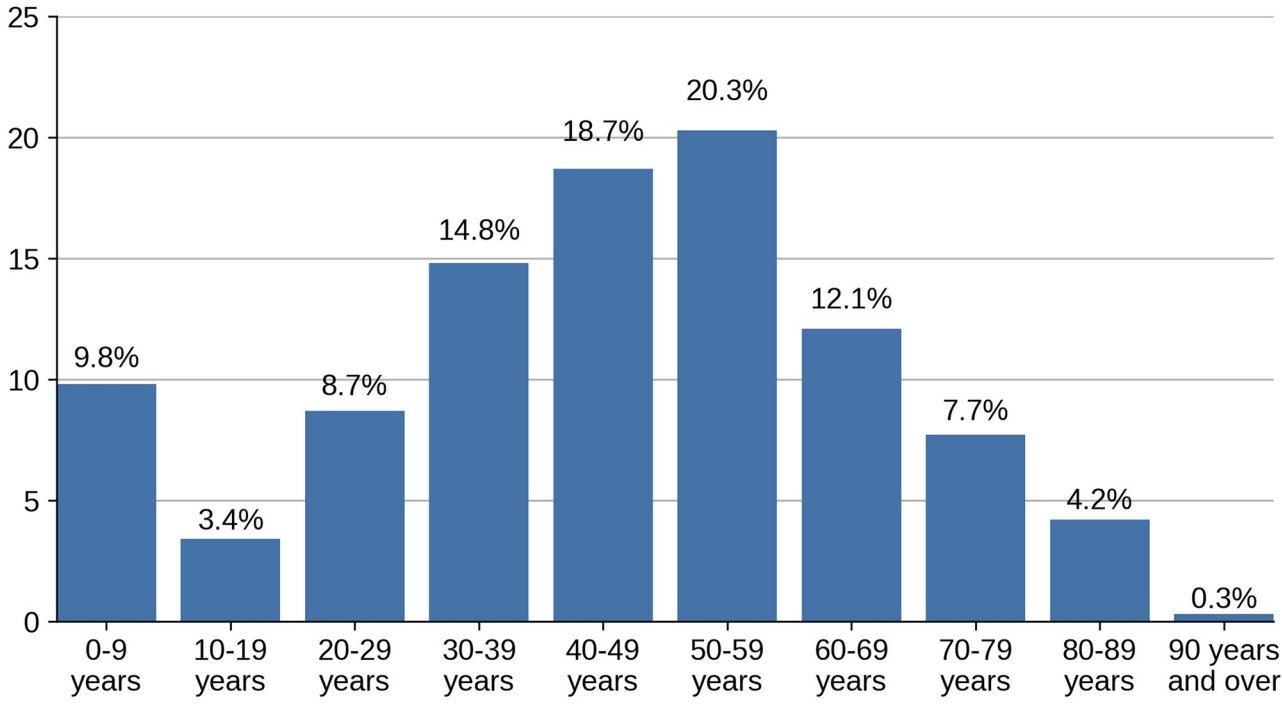

**Fig 1. Age profile of patients.**

**Table 1. Patient main presenting complaint.**

| Patient main presenting complaint | Count (n) | % |
|---|---|---|
| Musculoskeletal pain or dysfunction | 306 | 81.0 |
| Infancy-related complaints | 18 | 4.8 |
| Neurological | 16 | 4.2 |
| Other (see below) | 9 | 2.4 |
| Ear-nose-throat | 6 | 1.6 |
| Gastrointestinal | 5 | 1.3 |
| Psychological | 4 | 1.1 |
| Prevention/maintenance | 3 | 0.8 |
| Urogenital | 2 | 0.5 |
| Rheumatological | 2 | 0.5 |
| Cardiovascular | 2 | 0.5 |
| Respiratory | 1 | 0.3 |
| Obstetrical | 1 | 0.3 |
| General/non-specific | 1 | 0.3 |
| Endocrinological | 1 | 0.3 |
| Dentistry/orthodontics | 1 | 0.3 |
| Total | 378 | |

The "other" main presenting complaints, reported here verbatim, were: "reflux"; "overall wellbeing"; "nerve pain post shingles"; "clenching teeth"; "allergies"; "migraine"; "ME / CFS" (myalgic encaphalomyelitis/chronic fatigue syndrome); "checkup".

**Symptom duration.** The most common duration of symptoms for the presenting complaint before attending an appointment was 1–4 weeks (21.5%; 81/377), while 67.9% (256/377) of patients experienced persistent symptoms (13 weeks or longer), as shown in Fig 2.

## Consultation data

Just over half of treatment approaches used at first appointment and at second appointments comprised of soft tissue, articulatory, high-velocity low-amplitude (HVLA) thrust, stretching, and/or muscle energy techniques (51.6% and 56.5% respectively).

In almost half the recorded appointments osteopaths reported providing self-management advice and strategies (49.4%; 516/1,045). This comprised of stretching exercise, advice concerning physical activity, general physical activity, application of cold, and strengthening exercise.

Treatment approach data are presented in Table 2.

**Patient use of other healthcare modalities.** 36.6% of patients (138/377) had previous treatment or undergone investigations for the presenting episode, although only 14.8% (56/379) of patients were referred from another healthcare practitioner. Referrals were most frequently received from medical general practitioners (28.6%; 16/56). Full data are presented in Table 3.

**Patient symptoms.** 55.3% of symptoms reported by patients were of slow or insidious onset (208/376), 23.9% (90/376) were acute/ sudden (non-traumatic) and 17.6% (66/376) were from a traumatic onset. In 12 responses (3.2%) the onset was unknown or was not recorded. Five areas of the body accounted for 52.2% of predominant symptoms: lumbar spine (16.3%), neck (12.4%), shoulder (10.7%), thoracic spine (6.7%) and sacroiliac/pelvis/groin area (6.2%).

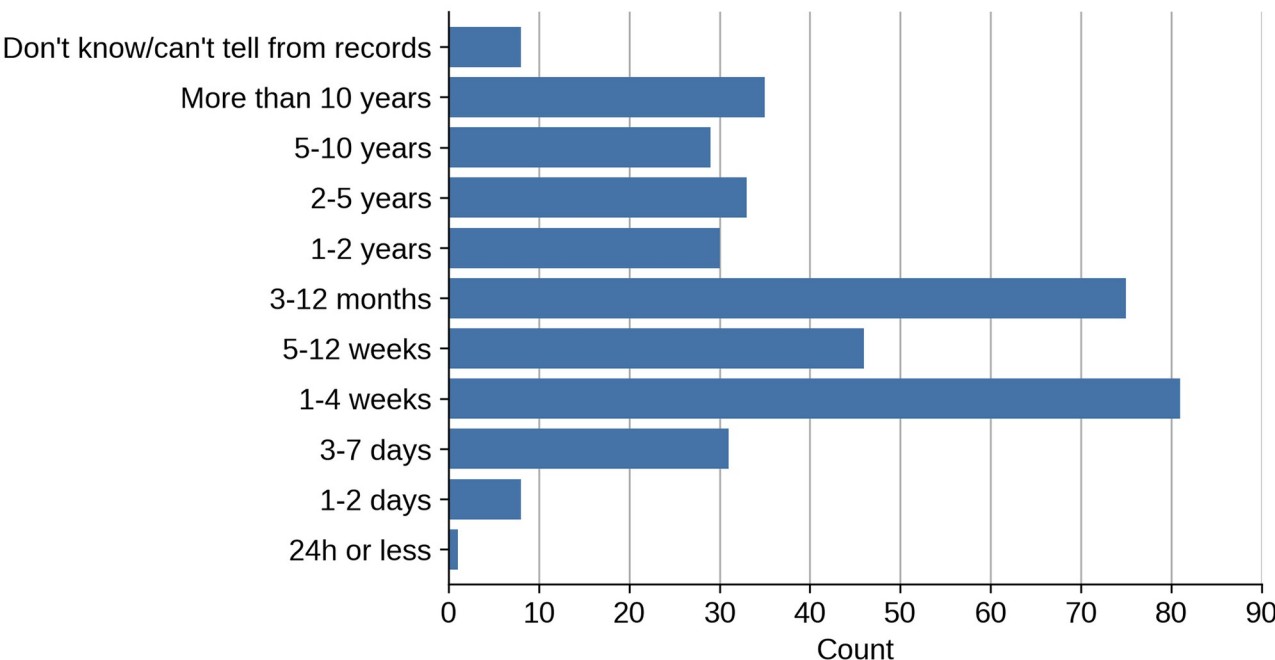

**Fig 2. For how long the patient had the presenting complaint before attending an appointment, including previous episodes.**

**Overall summary of findings.** The results showed that osteopaths work mostly on week-days and mostly alone, and are able to offer most patients an appointment within three days. Patients were most frequently in their mid-40s, and more females than males frequent osteo-pathic care. Most seek care for musculoskeletal conditions, notably long standing (more than 12 weeks of duration). The mode number of treatments per patient was four. Osteopaths treat their patients with a range of manual techniques including soft-tissue and articulatory tech-niques, and high velocity low amplitude thrust, but also provide support through advice and self-management guidance.

## Discussion

### Comparison with the survey of osteopaths in 2009

The OsteoSurvey 2019 employed REDCap survey software to support data collection online in contrast to the 2009 standardised data collection (SDC) study where osteopaths filled out paper questionnaires. This did not affect response rates: in 2009, 394 osteopaths responded (9.4% of the profession at the time) and 500 osteopaths responded in 2019 (9.4% of the profession).

In the decade between this survey and the last and earlier studies on profiles of osteopathic care, patient characteristics have remained broadly similar for adult age profiles, gender and presenting complaints [5, 17–21]. The presence of co-morbidities was also similar: in 2018, 41.7% of patients reported a range of comorbidities; within this number the most frequent were hypertension (11.0%), arthritis (8.3%), anxiety (5.9%), and asthma (5.1%). This profile is quite similar to that reported in 2009 where patients reported hypertension (11.7%), asthma (6.6%), arthritis (5.7%) and anxiety (3.6%) as the most frequent comorbidities. In the UK these issues represent 29.35% of total disease prevalence [22] which is broadly in line with the

**Table 2. Treatment approaches (first and second appointment).**

| Treatment approaches | 1st appt Count (n = 395) | % | 2nd appt Count (n = 395) | % |
|---|---|---|---|---|
| Soft tissue techniques | 292 | 73.9 | 243 | 61.5 |
| Articulatory techniques | 274 | 69.4 | 220 | 55.7 |
| HVLA thrust | 136 | 34.4 | 100 | 25.3 |
| Exercise—stretching | 130 | 32.9 | 79 | 20 |
| Muscle energy | 115 | 29.1 | 82 | 20.8 |
| Self-management | 106 | 26.8 | 58 | 14.7 |
| Cranial techniques | 91 | 23 | 77 | 19.5 |
| Lifestyle advice | 87 | 22 | 42 | 10.6 |
| Inhibition techniques (e.g. trigger points) | 75 | 19 | 46 | 11.7 |
| General osteopathic treatment (GOT) | 71 | 18 | 48 | 12.2 |
| Exercise—strengthening | 67 | 17 | 46 | 11.7 |
| Functional technique | 60 | 15.2 | 41 | 10.4 |
| Myofascial release (MFR) | 40 | 10.1 | 30 | 7.6 |
| Other | 36 | 9.1 | 12 | 3 |
| Relaxation | 34 | 8.6 | 15 | 3.8 |
| Biodynamic approach | 31 | 7.9 | 32 | 8.1 |
| Exercise—proprioception | 28 | 7.1 | 23 | 5.8 |
| Dry needling | 27 | 6.8 | 18 | 4.6 |
| Self-medication advice | 27 | 6.8 | 11 | 2.8 |
| Visceral | 20 | 5.1 | 11 | 2.8 |
| Strain/counterstrain | 18 | 4.6 | 16 | 4 |
| Dietary advice | 15 | 3.8 | 3 | 0.8 |
| Mindfulness | 15 | 3.8 | 4 | 1 |
| Pain neuroscience education (PNE) | 12 | 3 | 5 | 1.3 |
| Electro-therapy | 11 | 2.8 | 11 | 2.8 |
| Acupuncture | 4 | 1 | 2 | 0.5 |
| Psychological treatment | 3 | 0.8 | 1 | 0.3 |
| No hands-on treatment | 3 | 0.8 | 0 | 0 |
| Applied or clinical kinesiology | 2 | 0.5 | 3 | 0.8 |
| Orthotics | 2 | 0.5 | 2 | 0.5 |
| Nutrition therapy | 2 | 0.5 | 1 | 0.3 |

No responses were received for the following treatment options: injections; prescription of medication; bio-resonance therapy; herbal medicine; homeopathy; hypnosis.

numbers reported by osteopaths. The proportion of patients who had previously seen an osteopath remains at around 40%.

Management of symptoms in both surveys included a broad range of interventions used e.g. soft tissue techniques (78% in 2009 and 73.9% in 2018); articulation (72% in 2009 and 69.4% in 2018); HVLA thrust techniques (37.7% in 2009 and 34.4% in 2018), and cranial techniques (25.8% in 2009 and 23.0% in 2018). The data indicates that osteopaths may deliver 'packages of care' which include the promotion of self-management options such as education, advice, and exercise. This is in keeping with recommendations in current guidelines [3].

The costs of treatment were met by individuals in 88.4% of cases in 2018 compared with 89.1% in 2009. In 2019, 4.6% of patients had their treatment costs met by insurance schemes, 0.5% by their employer and 0.5% by the NHS. In 2009, 6.6% of patients had their treatment funded by health insurance schemes, 0.6% by their employer, and 0.6% by the NHS. We

**Table 3. Professions referring patients to osteopaths.**

| Professions referring patients to osteopaths | Count (n) | % |
|---|---|---|
| General practitioner | 16 | 28.6 |
| Complementary therapist | 12 | 21.4 |
| Another osteopath (including an assistant) | 10 | 17.9 |
| Another medical specialist | 7 | 12.5 |
| Physiotherapist | 4 | 7.1 |
| Midwife | 4 | 7.1 |
| Another allied health professional | 1 | 1.8 |
| Don't know/can't tell from records | 1 | 1.8 |
| Dentist | 1 | 1.8 |
| Total | 56 | |

There were 27 reports of osteopaths referring patients to other healthcare professions, with medical general practitioner again being the most common (55.6%; 15/27).

conclude that access to treatment not funded by individuals has remained static during the past decade. While there may be a variety of reasons for this, access to treatment still remains limited by ability to pay. Self-referral is still the most common route to treatment with 82.6% of patients being self-referred in 2018 compared with 79.9% in 2009, meaning that osteopaths work as first-contact practitioners.

However there are some changes. In the 2019 Osteosurvey, 13.2% (50/379) were under 20 years compared with 8.6% in 2009; and 4.7% were under 1 year in 2018 compared to 2.1% in 2009 suggesting an increase in consultations for much younger children.

Prior to attending an osteopathic practice in 2009, 48% of patients reported they had consulted their GP, compared with 41.3% (95/230) in 2018. In 2009, a total of 29% of patients had received previous treatment and investigations through the NHS including prescribed medication (20.1%), imaging (13.9%), hospital outpatient treatment (10.9%), and hospital inpatient treatment (1.3%). In contrast in 2018, 23.3% of patients reported undergoing imaging, and 6.1% had other forms of investigations including blood tests and urinalysis. The relationship between private and publicly funded care systems are closely linked for patients seeking osteopathic care.

Patients attending osteopathic practices in 2018 reported they experienced a range of symptoms including musculoskeletal pain and dysfunction (81.0%) and non-musculoskeletal symptoms (19.0%). In contrast, in 2009, 95.1% of patients reported musculoskeletal symptoms, with 4.3% of symptoms being non-musculoskeletal, indicating perhaps a greater diversity of care offered in 2018 and /or an increase in demand for non-musculoskeletal care.

Waiting time to access treatment has changed. In 2018, 48.6% of patients were seen within 3 days as opposed to 71% of patients in 2009. In 2009 there were around 6 osteopaths per 100,000 in the UK population and in 2018, 8 per100,000. Despite this apparent increase in osteopaths, quick access to an osteopath has fallen which may indicate increased demand or limited availability, for example indicated by the working hours mostly between 10.00am–4.00pm Monday to Friday.

In 2018, 67.9% of patients presented with persistent symptoms (13 weeks or longer). This included data concerning any previous symptom episode. In 2009 patients were asked about the duration of symptoms for their current episode which was 13 weeks or longer for 32.5% of patients. Both figures represent large numbers of patients with persistent symptoms, possibly reflecting the aging demographic in the UK population.

Other additional differences noted are changes to the management landscape. Osteopaths are implementing management approaches which indicate they may provide packages of care as recommended by clinical guidelines [3]. The clinician cohort who completed the survey may be more confident in their practices and motivated to engage in initiatives which demonstrate the full extent of osteopathic care.

## Comparison with other literature

**Osteopathy in other countries.** In a recent review of osteopathic care globally [23], the findings about osteopathic practice were similar at an international level. Osteopaths internationally work roughly the same number of hours per week, see similar numbers of patients per week, and osteopathic practitioners are most likely to work in one location and frequently on their own (with the exception of Italy). The vast majority of patients across the UK and central Europe are seen within one week. Musculoskeletal conditions (lower back and neck pain) account for the highest proportion of patient complaints across all countries. In central Europe the preferred techniques used by osteopaths were for the more gentle techniques such as osteopathy in the cranial field, visceral, functional and biodynamic techniques compared with the UK and Australian data that showed a preference towards more structural techniques such as soft tissue manipulation, articulation/mobilisation and spinal manipulation techniques. The UK compares with other countries showing that patients who most commonly attend osteopathic practitioners were employed/self-employed adults and more likely to be women than men.

**Comparison with other professions.** In the UK, physiotherapy and chiropractic are the predominant regulated physical therapies alongside osteopathy [24]. Comparison of patient and practise profiles within and across these professions is hindered by a lack of standardised approaches to data collection. This issue has been highlighted since at least 2012 when Moore, Bryant and Olivier [25] called for standardised data to reveal "who is accessing these services, who benefits from these services; how much these services cost in terms of clinicians time, the use of other healthcare resources and the effectiveness of interventions utilised in relation to quality outcomes". Systematic reviewers comparing the professions have echoed this, calling for researchers to "explore [the] characteristics of manual therapies" including "mode of administration, length of treatments, number of sessions, and choice of spinal region" [26]. The osteopathic profession appears to be alone in systematically collecting data that includes these characteristics. However, this standardised approach remains restricted to profession-wide surveys and has yet to become widely adopted in other research methodologies such as clinical trials.

## Strengths and limitations

Male participants were slightly under-represented in OsteoSurvey. In 2018 the GOsC register was comprised of 49% male osteopaths as opposed to the 41% male respondents in the survey, and when we compared the age profile of registered osteopaths and responders, osteopaths between the ages of 26 and 35 years were also under-represented. While we cannot be certain, it seems unlikely that these under-representations have introduced significant biases into the data since our findings have remained broadly consistent with the previous 2009 survey.

The response rate for this survey was not as high as we would have liked but the overall sample size for patients was sufficient for our analysis. We chose a retrospective audit of patient records which may have proved difficult for some clinicians as their records may not have contained the necessary information to complete the questionnaire. However, we thought this may be a finding in itself to highlight areas where record keeping could be improved.

Overall the amount of missing data did not highlight any particular area of poor record keeping.

We have been able to compare some data from the 2009 and 2019 surveys, and this is the first assessment of this nature within the profession to describe any change over time.

## Future research

Other surveys suggest that awareness of osteopathy by the population remains low. An independent survey conducted by YouGov indicated that 57% of people who had not seen an osteopath wanted assurances on a recognised level of education and training, 65% expected good quality advice and treatment, and 90% wanted evidence of effectiveness or recommendation [27]. After the 2009 survey a recommendation was made for the profession to develop a system for independent outcome data collection. This has resulted in the development of the Patient Reported Outcome Measurement (PROMs) system. This system has collected some encouraging outcome data collected directly from patients and independent to the clinician delivering care [28]. Promoting the findings of the PROMs data and information concerning clinicians from the OsteoSurvey 2019 study will start to fill the information gap identified by patients.

Profiling osteopaths, their patients and the nature and type of care helps to describe the profession which is useful for providing information for the profession, its regulatory body, its education institutions and its professional body and for informing other health care practitioners about osteopathy. However, more data is needed about patients, to understand their expectations, experiences and outcomes this information would enable practitioners and the profession as a whole to reflect on the nature and type of care they give and its impact on patients.

## Implications for the profession

The future of the UK osteopathic profession will depend on its ability to adapt to the changing health care needs of the nation. Traditionally osteopaths have filled these gaps for example for persistent pain and other conditions not well managed within the NHS or by pharmaceuticals. There is some indication of flexibility and adaptability which could be enhanced through education, training and active marketing to reflect demographic changes and areas where health service provision is not meeting demand. As the aging UK population grows, demand for care for persistent musculoskeletal conditions and other age related disorders will increase, for osteopathy to maintain and sustain its presence it will need to ensure it offers patients a unique experience and health and wellbeing benefit.

## Conclusion

Osteopaths provide a variety of approaches to the care and management of a range of conditions that are predominantly musculoskeletal. In comparison to the 2009 data, osteopathic patient profiles in the UK remain largely unchanged over the past decade. Notable differences include increases in: the number of younger children treated by osteopaths; waiting times for osteopathic appointments; and the proportion of patients presenting with persistent symptoms. There was a reduction in the number of patients who had previously consulted their GP or received investigations or treatment via the NHS.

Our study, and the 2009 study preceding it, provide insight into the patient and practise profiles of the osteopathic profession in the UK, and contribute to similar data for the osteopathic professions in a growing range of countries. Comparable data for the physiotherapy and chiropractic professions in the UK appears rare, suggesting that this may be difficult to obtain or of less interest to those professions. There are some indications that the osteopathic

profession may be responsive to expectations that clinicians provide a 'package of care' which is based on current guidelines and in keeping with wider healthcare practises. To better understand the role of osteopathy in UK health service delivery, the profession needs to do more research with patients in order to understand their needs and their expected outcomes of care, and for this to inform osteopathic practice and education.

## Supporting information

**S1 File. Questionnaire survey content.**
(DOCX)

## Acknowledgments

The working group: Steve Vogel, Martin Pendry, Phil Bright, Maria Fitzgerald.

## Author Contributions

**Conceptualization:** Carol Fawkes, Dawn Carnes.

**Data curation:** Austin Plunkett.

**Formal analysis:** Austin Plunkett, Carol Fawkes.

**Funding acquisition:** Dawn Carnes.

**Investigation:** Austin Plunkett.

**Methodology:** Carol Fawkes, Dawn Carnes.

**Project administration:** Austin Plunkett.

**Supervision:** Dawn Carnes.

**Visualization:** Austin Plunkett.

**Writing – original draft:** Austin Plunkett, Carol Fawkes, Dawn Carnes.

**Writing – review & editing:** Austin Plunkett, Carol Fawkes, Dawn Carnes.

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
