## [Decision Letter · Decision Letter 0]

13 Sep 2021

PONE-D-20-30382UK osteopathic practice in 2019: a retrospective analysis of practice data.PLOS ONE

Dear Dr. Plunkett,

Thank you for submitting your manuscript to PLOS ONE. After careful consideration, we feel that it has merit but does not fully meet PLOS ONE’s publication criteria as it currently stands. Therefore, we invite you to submit a revised version of the manuscript that addresses the points raised during the review process.

 The reviewers were positive about the contribution your work represents, but also identified several ways in which the interpretation of the results could be strengthened and additional context provided in the Discussion. Please respond carefully to each of their suggestions when preparing your revisions.

We look forward to receiving your revised manuscript.

Kind regards,

Jamie Males

Staff Editor

PLOS ONE

Journal Requirements:

"AP is a Director at the Institute of Osteopathy, the members' organisation for the UK's osteopathic profession."

6. Please note that in order to use the direct billing option the corresponding author must be affiliated with the chosen institute. Please either amend your manuscript to change the affiliation or corresponding author, or email us at plosone@plos.org with a request to remove this option.

7. Please ensure that you refer to Figures 1 and 2 in your text as, if accepted, production will need this reference to link the reader to the figure.

8. We note you have included a table to which you do not refer in the text of your manuscript. Please ensure that you refer to Tables 1,2 and 3 in your text; if accepted, production will need this reference to link the reader to the Table.

Reviewers' comments:

Reviewer's Responses to Questions

**Comments to the Author**

1. Is the manuscript technically sound, and do the data support the conclusions?

Reviewer #1: Partly

Reviewer #2: No

2. Has the statistical analysis been performed appropriately and rigorously? 

Reviewer #1: Yes

Reviewer #2: Yes

3. Have the authors made all data underlying the findings in their manuscript fully available?

Reviewer #1: Yes

Reviewer #2: No

4. Is the manuscript presented in an intelligible fashion and written in standard English?

Reviewer #1: Yes

Reviewer #2: Yes

5. Review Comments to the Author

Reviewer #1: The article untitled UK osteopathic practice in 2019: a retrospective analysis of practice data presents the results of a web-based system survey about osteopathic practice in UK.

General comments:

The article is relevant and of interest. The design is appropriate. Data collection is well done. In the results section, it would be really more interesting to have a comparison between the 2019 and 2009 results (same authors?). The discussion deserves to be reoriented to give less numbers and more explanation of changing trends and potential causes. The conclusions are also to be reoriented in relation to the results.

Abstract

The conclusion in the abstract is not related directly to results presented (p. 3 lines 43-45).

Introduction and aim:

The context for this update is well defined. It would be interesting to mentioned new potential roles expected for this profession in the introduction. The aim is explicit and well detailed.

Methods:

Page 4, line 82: remove coma.

Page 5, lines 96-97: The survey questions were mainly derived from the original standardised data collection survey conducted in 2009 [4] and from a survey commissioned by the GOsC [14]. What changes have been made to the original tool? Could this change the content validity?

Results

Table 2 could be shorter and mention that some approaches have not been reported by any osteopath (page 13).

It would be really interesting if comparisons could be made between 2019 and 2009 directly in the results section including in the tables with a notion of statistically significant difference or not. This would allow further discussion of major changes over time and possible reasons for these changes.

Discussion

Not sure if it is relevant to repeat the statistics at the beginning of the discussion. Maybe just the broad findings (page 15 lines 260-273).

Standardize references including surveys previously conducted in the UK (page 16, lines 281-282).

No need to describe the percentage of co-morbidities again. More emphasis on comparison with 2009 (page 16, lines 283-286).

The comparison with other countries is very similar to the discussion in the systematic review. Possibility to put more emphasis on what is different between the current study and the systematic review cited (page 18).

Strengths and limitations

What might be the implication of under-representation of certain age or gender groups?

Conclusions

The conclusion is more of a perspective than a conclusion related to the results or evolution of the osteopaths' practice.

Reviewer #2: Thank you for the opportunity to review this manuscript. Overall this is an interesting contribution to the literature. I have made comments and suggestions throughout the attached document. However, the major considerations centre on how the results of the study are contextualised to the wider literature - the reader is left to ponder the value of the data in the study. At present, the Discussion is largely a descriptive comparison to 2009 work with little reference to other works to explain why changes were observed or not observed. If the focus is to solely be a comparison between 2009 and 2019, this could be reflected in the title of the paper. Otherwise greater attention should be given to discussing the results in the context of other literature.

6. PLOS authors have the option to publish the peer review history of their article (what does this mean?). If published, this will include your full peer review and any attached files.

Reviewer #1: No

Reviewer #2: **Yes: **Brett Vaughan

---

## [Author Response · Author response to Decision Letter 0]

12 Nov 2021

Please see highlighted edits in the latest submission, and comments in the document "response-to-reviewers.docx".

---

## [Decision Letter · Decision Letter 1]

3 Feb 2022

PONE-D-20-30382R1Osteopathic practice in the United Kingdom: a retrospective analysis of practice dataPLOS ONE

Dear authors

Thank you for submitting your manuscript to PLOS ONE. After careful consideration, we feel that it has merit but does not fully meet PLOS ONE’s publication criteria as it currently stands. Therefore, we invite you to submit a revised version of the manuscript that addresses the points raised during the review process.

The discussion and conclusion need to be highlighted with the salient findings so that the readers can easily understand the importance of study in UK

We look forward to receiving your revised manuscript.

Kind regards,

Narasimha Murthy Bhamidipati, Ph.D

Academic Editor

PLOS ONE

---

## [Author Response · Author response to Decision Letter 1]

11 Mar 2022

We have provided a detailed response in the document "plosone-response-to-reviewer-final.docx" in this latest submission.

---

## [Decision Letter · Decision Letter 2]

20 Jun 2022

Osteopathic practice in the United Kingdom: a retrospective analysis of practice data

PONE-D-20-30382R2

Dear Dr. Plunkett,

We’re pleased to inform you that your manuscript has been judged scientifically suitable for publication and will be formally accepted for publication once it meets all outstanding technical requirements.

Kind regards,

Emiliano Cè

Academic Editor

PLOS ONE

Additional Editor Comments (optional):

Reviewers' comments:

Reviewer's Responses to Questions

**Comments to the Author**

1. If the authors have adequately addressed your comments raised in a previous round of review and you feel that this manuscript is now acceptable for publication, you may indicate that here to bypass the “Comments to the Author” section, enter your conflict of interest statement in the “Confidential to Editor” section, and submit your "Accept" recommendation.

Reviewer #1: All comments have been addressed

2. Is the manuscript technically sound, and do the data support the conclusions?

Reviewer #1: Yes

3. Has the statistical analysis been performed appropriately and rigorously? 

Reviewer #1: Yes

4. Have the authors made all data underlying the findings in their manuscript fully available?

Reviewer #1: Yes

5. Is the manuscript presented in an intelligible fashion and written in standard English?

Reviewer #1: Yes

6. Review Comments to the Author

Reviewer #1: (No Response)

7. PLOS authors have the option to publish the peer review history of their article (what does this mean?). If published, this will include your full peer review and any attached files.

Reviewer #1: No

---

## [Editor Report · Acceptance letter]

24 Jun 2022

PONE-D-20-30382R2 

Osteopathic practice in the United Kingdom: a retrospective analysis of practice data 

Dear Dr. Plunkett:

I'm pleased to inform you that your manuscript has been deemed suitable for publication in PLOS ONE. Congratulations! Your manuscript is now with our production department. 

Kind regards, 

on behalf of

Professor Emiliano Cè 

Academic Editor

PLOS ONE